# Development of Clinical Radiomics-Based Models to Predict Survival Outcome in Pancreatic Ductal Adenocarcinoma: A Multicenter Retrospective Study

**DOI:** 10.3390/diagnostics14070712

**Published:** 2024-03-28

**Authors:** Ayoub Mokhtari, Roberto Casale, Zohaib Salahuddin, Zelda Paquier, Thomas Guiot, Henry C. Woodruff, Philippe Lambin, Jean-Luc Van Laethem, Alain Hendlisz, Maria Antonietta Bali

**Affiliations:** 1Radiology Department, Institut Jules Bordet Hôpital Universitaire de Bruxelles, Université Libre de Bruxelles, 1070 Brussels, Belgium; 2Department of Precision Medicine, GROW—Research Institute for Oncology and Reproduction, Maastricht University, 6220MD Maastricht, The Netherlands; 3Medical Physics Department, Institut Jules Bordet Hôpital Universitaire de Bruxelles, Université Libre de Bruxelles, 1070 Brussels, Belgium; 4Department of Radiology and Nuclear Medicine, GROW—School for Oncology and Reproduction, Maastricht University Medical Centre+, 6229HX Maastricht, The Netherlands; 5Department of Gastroenterology and Digestive Oncology, Hôpital Universitaire de Bruxelles, Université Libre de Bruxelles, 1070 Brussels, Belgium

**Keywords:** radiomics, computed tomography (CT), pancreas, pancreatic ductal carcinomas, survival analyses

## Abstract

**Highlights:**

**What are the main findings?**

**What is the implication of the main finding?**

**Abstract:**

Purpose. This multicenter retrospective study aims to identify reliable clinical and radiomic features to build machine learning models that predict progression-free survival (PFS) and overall survival (OS) in pancreatic ductal adenocarcinoma (PDAC) patients. Methods. Between 2010 and 2020 pre-treatment contrast-enhanced CT scans of 287 pathology-confirmed PDAC patients from two sites of the Hopital Universitaire de Bruxelles (HUB) and from 47 hospitals within the HUB network were retrospectively analysed. Demographic, clinical, and survival data were also collected. Gross tumour volume (GTV) and non-tumoral pancreas (RPV) were semi-manually segmented and radiomics features were extracted. Patients from two HUB sites comprised the training dataset, while those from the remaining 47 hospitals of the HUB network constituted the testing dataset. A three-step method was used for feature selection. Based on the GradientBoostingSurvivalAnalysis classifier, different machine learning models were trained and tested to predict OS and PFS. Model performances were assessed using the C-index and Kaplan–Meier curves. SHAP analysis was applied to allow for post hoc interpretability. Results. A total of 107 radiomics features were extracted from each of the GTV and RPV. Fourteen subgroups of features were selected: clinical, GTV, RPV, clinical & GTV, clinical & GTV & RPV, GTV-volume and RPV-volume both for OS and PFS. Subsequently, 14 Gradient Boosting Survival Analysis models were trained and tested. In the testing dataset, the clinical & GTV model demonstrated the highest performance for OS (C-index: 0.72) among all other models, while for PFS, the clinical model exhibited a superior performance (C-index: 0.70). Conclusions. An integrated approach, combining clinical and radiomics features, excels in predicting OS, whereas clinical features demonstrate strong performance in PFS prediction.

## 1. Introduction

Pancreatic ductal adenocarcinoma (PDAC) is one of the most aggressive malignancies. Its incidence continues to rise, but it maintains a low 5-years survival rate, below 12% [1]. The poor prognosis is a result of a lack of early screening biomarkers, late diagnosis, early metastatic dissemination, and tumour resistance to systemic therapies. Surgery remains the only treatment with curative intent but less than 20% of the patients are candidates at the time of the diagnosis.

Differences in response to treatment and in clinical outcomes have been related to PDAC high molecular heterogeneity [2] and to the presence of a predominant stromal component comprising up to 90% of the total tumour volume [3]. Based on gene expressions and stromal characteristics, several subtypes of PDAC have been identified which differ in their clinical behaviour and response to treatment [4]. 

Inter-patient heterogeneity is a key problem that has led to the failure of many clinical trials, highlighting the need for a more stratified therapeutic approach based on PDAC taxonomy. However, pragmatic approaches to subtyping for clinical implementation and survival prediction are lacking.

The answer may come from the emerging field of radiomics where image-derived features and radiomics features are linked to genomic profiles and beyond. This process, based on sophisticated statistical and machine learning approaches, may provide non-invasively important tumour characteristics, reflecting tumour biology with a major impact on the optimization and personalization of therapeutic strategies, on individual patients’ benefit and on overall disease survival [5]. The extraction of these quantitative radiomics features can be correlated with different clinical outcomes, highlighting information that is not visible to the human eye which can be obtained from different imaging methods [6,7]. Radiomics features can be extracted from conventional imaging modalities and provide information about cancer biology; for example, tumour heterogeneity, which can be manifested at different levels (phenotypic, genomic, …) and reflects tumour aggressiveness, tumour grade, clinical outcomes, response to treatment and survival outcomes [7,8,9,10,11].

The usefulness of this new approach has been reported in several cancer types, including PDAC, for predicting response to treatment and survival outcomes [11]. More recent results of a radiogenomic approach for PDAC have demonstrated the possibility of identifying features correlated with gene mutations (mainly SMAD4, which could be associated with disease-free survival) and stromal content [12,13,14,15]. Both genetic mutations and stromal components have been considered to be predictive of response to treatment [2,3,15]. Other studies have also demonstrated the role of radiomics in characterising PDAC and predicting resectability, risk of recurrence and overall survival [6,11].

However, previous studies in this area have often involved small patient cohorts with limited external validation, leading to findings that may not be generalizable. Notably, multicentric studies in this specific research domain are scarce, which restricts the dependability of the established knowledge [16,17,18,19]. In response to these gaps, the present study aims to identify predictive biomarkers for overall survival (OS) and progression-free survival (PFS) in patients with PDAC, using the radiomics-based approach of pre-treatment computed tomography (CT) images and clinical data. Our non-invasive approach, which synergizes medical imaging with machine learning, holds promise for improving patient management.

A key contribution and innovation of this study is its multicentric design, involving data from a network of 47 different hospitals, enabling a comprehensive evaluation of the proposed models’ effectiveness in a clinical routine. This large-scale, real-world scenario assessment underscores the practical utility of our methodology across diverse clinical settings. Moreover, the study delves into the significance of both clinical and radiomic features in the predictive models, providing insights into their utility in personalised care strategies based on survival outcomes prediction.

The structure of this article is organised to provide a comprehensive exploration of the study. Following the “Introduction”, which sets the stage by introducing the topic and outlining the rationale of our study, the manuscript is divided into several key sections. The “Materials and Methods” section elaborates on the methodology employed, detailing the patient selection process, the clinical data included in the study and the radiomic procedure—from image selection to the creation and testing of models—as well as the statistical analyses undertaken. The “Results” section presents the findings, focusing on the performance of the models, model explainability, and the survival curves of identified groups with varying risk levels. Finally, the “Discussion” section reflects on these results, comparing them with relevant literature, and addresses the limitations of the current study.

## 2. Material and Methods 

### 2.1. Study Population

This retrospective study has been approved by the ethical review boards of the Hopital Universitaire de Bruxelles (HUB), respectively, of the Hopital Erasme (HE) site and of the Institut Jules Bordet (IJB) site and patient written consent was waived.

Between January 2010 and December 2020, patients with histologically proven PDAC were identified from HE (dataset 1), from IJB (dataset 2) and from the remaining hospital network connected to HUB (dataset 3—images stored in HUB archive system).

Inclusion criteria were: histologically proven PDAC (based on endoscopic-driven biopsy), patients older than 18 years, pre-treatment contrast-enhanced CT at the portal phase with a slice thickness greater than 1 mm and with clearly visible PDAC lesion. Exclusion criteria were: patients with no pretreatment CT, or with a pretreatment CT without portal phase or with biliary stent at the moment of CT scanner and patients with final diagnosis other than PDAC.

For each included patient the following clinical, histological and radiological data, representing the clinical features, were collected: age, gender, body mass index (BMI), alcohol consumption, tobacco consumption, Ca19.9, treatment type, treatment strategy, histological grade, resection margin, pancreatic tumour location, presence/absence of metastatic disease, disease subgrouping based on NCCN guidelines (version 2.2022 [20,21]). 

OS was calculated in months from the date of diagnosis to date of death and PFS was calculated in months from the date of treatment initiation to the date of first progression. For PFS and OS, patients were considered to be censored if they were lost during follow-up. Additional details can be found in the Appendix A “Clinical, radiological and histological data”. If clinical features were missing, data were filled using the MissForest package version 0.2.0 (https://pypi.org/project/MissForest/ (accessed on 13 May 2023)).

### 2.2. Patients Stratification

The exams in dataset 1 and dataset 2 were used as the training and validation dataset, specifically for feature selection and model evaluation through ten-fold cross-validation. Exams from dataset 3, comprising data from 47 different hospitals with varying CT acquisition protocols, were used as the testing dataset, which remained separate throughout the feature selection and model training phases. This stratification was used for both outcomes, OS and PFS.

### 2.3. Data Analysis 

All patients underwent abdominal pretreatment contrast-enhanced CT. All the DICOM data were stored in the radiological department server of IJB, in accordance with the approbation of the ethical review boards. The portal phase was taken into account for this study. 

Subsequently, radiomics analysis was applied to the acquired images. This process involved: segmentation of the regions of interest, extracting quantitative features from the CT images (which capture the texture, shape, and intensity of the tumour region), feature selection to identify the most relevant features for our predictive models. These radiomic features were then used in conjunction with the clinical data to develop the predictive models for PFS and OS in PDAC patients. Further details are provided in the following paragraphs.

#### 2.3.1. Segmentation and Features Extraction

Contrast-enhanced CT images were anonymized, and portal imaging sequences were sent to an archive connected with MIM 7.1.5™ (MIM Software Inc., Cleveland, OH, USA).

The CT images had different resolutions; therefore, all the exams were resampled to the same resolution, using the SimpleITK sitkBSpline interpolator for exams and the SimpleITK sitkNearestNeighbor interpolator for segmentations.

A radiologist with 12 years’ experience in abdominal imaging (RC, Reader 1) delineated and segmented semi-manually each tumour using MIM 7.1.5™ to obtain gross tumour volume (GTV1) (Figure 1). To assess the intra-rater reliability of features, Reader 1 performed the tumour segmentations a second time (GTV2) in 45 randomly chosen patients after a few months. To assess the inter-rater reliability of features for GTV, a first-year radiologist resident (AM, Reader 2) delineated and segmented the pancreatic tumours from the same 45 randomly chosen patients (GTV3). Intra-rater and inter-observer reliability evaluation of features was obtained by using the IntraClass Correlation two-way mixed effect single measurement (ICC2). Only features with ICC2 greater than 0.75 were selected [22,23].

An automated segmentation tool developed at Maastricht University was used to delineate and automatically segment the non-tumoral pancreas (RPV) [24]; subsequently, the RPV segmentations were manually verified and modified, using ITK-SNAP [25] (Version 3.8.0, http://www.itksnap.org/ (accessed on 13 May 2023)) (an example is provided in Appendix A) by the two readers in consensus. 

Radiomics features were extracted from GTV1, GTV2, GTV3 and RPV segmentations with PyRadiomics [26]. For each segmentation, a total of 107 distinct features were extracted, encompassing seven primary groups: shape (14 features), first order (18 features), GLCM (24 features), GLDM (14 features), GLRLM (16 features), GLSZM (16 features) and NGTDM (5 features). Definitions of extracted features are described at https://pyradiomics.readthedocs.io/en/latest/features.html (accessed on 13 May 2023). To assure better reproducibility, the intensity discretisation bin width was set to 25 [27].

#### 2.3.2. Feature Selection

Clinical features and features obtained from segmentations (GTV1, RPV) were used individually and in combination and the following feature subsets were obtained: clinical, GTV1, RPV, clinical&GTV1, clinical&GTV1&RPV, for both OS and PFS, resulting in 10 feature groups.

Initially, the features with the highest reliability based on the aforementioned ICC2 cutoff were selected (this step was applied only to GTV1 features).

Subsequently, for each group, highly correlated and redundant features were eliminated by applying a Spearman correlation coefficient threshold of 0.80. 

Then, a univariate analysis was conducted to identify the most significant features that exhibit the highest C-index within Cox’s Proportional Hazards [28,29].

Lastly, for every group and using the previous assessment of the single features’ importance, the features were progressively combined to determine the most optimal combination, employing a Cox Proportional Hazards model.

Further details regarding the feature selection process are described in the GitHub repository: https://github.com/roberto-casale/RadPanc-clinical-radiomics-based-models (accessed on 13 January 2024). 

In addition, as distinct sets, we used the volume measure for GTV1 and RPV, both for OS and PFS, resulting in a total of 14 sets of features (Figure 2).

It is worth mentioning that these steps were computed only on the training–validation dataset (dataset 1 and dataset 2).

### 2.4. Model Building and Evaluation

Using the GradientBoostingSurvivalAnalysis classifier (scikit-survival version 0.19.0.post1) [28], ten-fold cross-validation was performed on the training–validation dataset to tune the hyperparameter and for evaluation. Fourteen different models were trained, in particular they were based on each set of features, both for OS and PFS and tested on the testing dataset.

### 2.5. Model Interpretability

To explain the model’s prediction, we used SHapley Additive exPlanations (SHAP), a post hoc interpretability technique that assigns a SHAP value to each feature [30,31]. The SHAP value of each feature quantifies its impact on the model’s output. Positive SHAP values show that the presence of a feature pushes the prediction higher than the baseline and negative values suggest the opposite. The magnitude of these values demonstrates the strength of the feature’s influence on the model’s output. To provide a more comprehensive understanding, we used SHAP global summary plots. A SHAP summary plot ranks the features based on their average absolute SHAP values across all predictions and it helps to visualise their overall importance. Hence, SHAP helps to identify the most influential features and their trends in our best models for predicting OS and PFS.

### 2.6. Statistical Analysis

Chi-squared and Mann–Whitney were used to compare the clinical features between the training–validation and testing datasets. To analyse the various models, including those based solely on volume, we employed the C-index. This index generalises the area under the receiver operator characteristic curve (AUC) by assessing how well the model can separate the survival curves [32]. 

A two-sided permutation test was used to assess the statistical significance of the C-index for survival prediction across the different models in the testing dataset.

For survival prediction, the top-performing models that predicted OS and PFS were analysed and used to obtain the risk score through the GradientBoostingSurvivalAnalysis model. For the classification of high and low-risk groups, we used the median risk score from the training–validation dataset as a threshold [33]. Specifically, patients in the test set with risk scores above this median were classified as high risk, while those with scores below the median were categorised as low risk.

Subsequently, survival curves were generated using the Kaplan–Meier method to showcase the survival probabilities over time for these two groups. To assess the statistical difference between the low-risk group curve and high-risk group curve, in the testing dataset, the two-log-rank test was used.

The Spearman correlation coefficient was used to calculate the inter-correlation among the selected features and with volume. 

All the above mentioned parts of the pipeline were computed using Python 3.8; the features were standardised using sklearn.preprocessing.StandardScaler (scikit-learn version 1.1.2) [34].

## 3. Results 

### 3.1. Clinical Data and Patient Stratification

The total patient population from the three datasets was 1040. In total, 753/1040 (72.4%) patients were excluded (Figure 3). The final study population consisted of 287/1040 patients (27.6%). The median OS was 9 months (range 0 to 63) and the median PFS was 3 months (range 0 to 35). Of the patients, 37/287 (12.9%) were censored for OS and 77/287 (26.8%) were censored for PFS because of a lack of follow-up. 

The study population was categorised into a training–validation group (only patients from dataset 1 and dataset 2) and a testing group (only patients from dataset 3). The clinical data from the training–validation datasets and test datasets are presented in Table 1; additional details regarding ordinal encoding for clinical data can be found in the Appendix A“Ordinal encoding for clinical data”. 

### 3.2. Image Acquisition and Segmentation

CT exams were obtained from 24 different scanners, from 49 different centres (more details in Appendix A); CTscanners included 16-slice, 64-slice and 128-slice scanners. Slice thicknesses ranged from 1 to 5 mm; the median value of the resolution was 0.73 × 0.73 × 1.5 mm^3^ in the whole dataset and it was 0.72 × 0.72 × 1.5 mm^3^ in the training–validation dataset.

Before feature extraction, all the data were resampled to a resolution of 0.72 × 0.72 × 1 mm^3^.

### 3.3. Features Selection

A total of 107 features were extracted separately from GTV1, GTV2, GTV3 and RPV. There were no missing radiomics feature values after feature extraction.

For the GTV1 segmentations, the median volume was 11.7 cm^3^ (range 0.8–225.4 cm^3^); for the RPV segmentations, the median volume was 65.2 cm^3^ (range 8.3–151.0 cm^3^).

After assessing the intra-rater and inter-observer reliability, 63/107 features were selected for GTV1 (more details in Appendix A “Assessment of feature repeatability”). The evaluation and removal of highly correlated features is shown in Appendix A “Independent features”. Finally, after the univariate analysis, using a Cox proportional hazards model, and the search for the best combination, 10 different sets of features were obtained for the following subsets: clinical, GTV1, RPV, clinical&GTV1 and clinical&GTV1&RPV, for both OS and PFS (Table 2 and Table 3). 

The intercorrelation among the selected features, including volumes, was assessed in the testing dataset (Appendix A).

### 3.4. Model Performances

Using the GradientBoostingSurvivalAnalysis classifier (setting parameters are reported in Appendix A), fourteen different models were built (seven for predicting OS and seven for predicting PFS). A C-index was computed for each model trained on the training–validation dataset using ten-fold cross-validation and on the testing dataset using models trained on the training–validation dataset (results are presented in Table 4 and Table 5 and in Appendix A). 

The models that relied solely on volume as a predictive factor exhibited poor performances (Appendix A).

The results of the univariate analysis for each individual clinical and radiomic feature used in the models are presented in the Appendix A “Univariate analysis”.

The results pertaining to the c-index values for the prediction of OS and PFS, using the best-performing models within distinct patient subgroups undergoing different types of treatment, are available in the Appendix A “C-Indexes across different types of treatment”.

### 3.5. Explainability of the Models

In Figure 4 and Figure 5, the colour coding is used to represent the values of the features for the best-performing models. Specifically, a red colour indicates high feature values, while a blue colour signifies low feature values. According to the C-index, the Clinical&GTV1 model was the best model for OS prediction; for the subgroups CA19, R_Margins, Age and Grading, higher feature values were found to be positively correlated with a higher risk score, indicating a poorer prognosis (Figure 4). Conversely, first_order_90Percentile (from GTV1) exhibited an inverse relationship, with lower values being associated with a higher risk score. It is worth noting the highest value for R_Margins meant that patients did not undergo surgery.

According to the C-index, the clinical model was the best model for PFS prediction; the SHAP summary plot (Figure 5) showed that the subgroups R_Margins, Tobacco, Localization, CA19 and Grading demonstrated higher feature values related to higher risk scores (the highest value for Localisation meant tumours located at the tail); while age and alcohol consumption highlighted that the lowest value was associated with a higher risk score.

### 3.6. Survival Prediction

For OS and PFS prediction, we partitioned the testing dataset into two risk groups. Figure 6 and Figure 7 illustrate Kaplan–Meier survival plots and counts indicating the respective quantities of patients who were at risk, censored and observed at various time points in the survival plots. According to the two log-rank tests, the high-risk group exhibited significantly inferior survival compared to the low-risk group (*p* < 0.005) for both PFS and OS. 

## 4. Discussion

In this study, a comparative evaluation was performed on various models that integrated either or both clinical and radiomics features extracted from pretreatment CT scans to predict OS and PFS. Our findings indicate that the combination of clinical and GTV1 radiomics features yielded the best predictive model for OS, achieving a C-index of 0.72 in the testing dataset. Conversely, for PFS prediction, the best model was obtained using only clinical features, with a C-index of 0.70 for the testing dataset. The efficacy of the alternative models was relatively inferior in comparison.

Examining the statistical comparison among the different models, it is evident that models relying solely on clinical features showcase excellent performance in predicting both OS and PFS. While the Clinical&GTV1 model slightly surpasses the Clinical model in terms of C-index for OS, the difference between the two models lacks statistical significance (*p*-value = 0.212; Appendix A). The analysis underscores the robustness of clinical-based models in prognostication for OS and PFS.

From another perspective, in the absence of clinical data, the radiomic model based solely on GTV1 can be a viable alternative for predicting OS, as no statistically significant difference was observed between the Clinical and GTV1 models (*p*-value = 0.156; Appendix A). Regarding PFS, however, the Clinical model demonstrates a statistically superior C-index compared to the other models based solely on radiomic features.

Examining the Kaplan–Meier curves, the risk score derived from the top-performing models can be utilised to categorise patients into high-risk and low-risk categories for both OS (Figure 6) and PFS (Figure 7).

Our findings are consistent with the recent literature. For predicting survival time after the surgical resection, Park et al. [35] reported a C-index of 0.74 for their clinical and radiomics model; differently from our results, they demonstrated a slight increase in the C-index when adding RPV features to the clinical + GTV model. Cheng et al. [16] observed that CT texture analysis was associated with PFS and OS; by combining texture features and tumour size, they achieved an AUC of 0.756 for predicting OS. In contrast to our findings, CA19.9 levels were not found to be correlated with either OS or PFS. This discrepancy in the importance of CA19.9 might be attributed to the relatively small dataset, which consisted exclusively of 41 patients with unresectable PDAC. Separate from our findings, both previous studies [16,35] did not investigate the inter- or intra-observer reliability of segmentations. In a study by Yang et al. [36], a radiomics signature was proposed to predict early death (within one year) in patients with advanced PDAC exhibiting stable disease. The radiomics signature achieved an AUC of 0.84 in the internal testing set and 0.87 in the external testing set. However, their inclusion criteria focused on pancreatic tumours with a size greater than or equal to 20 mm and patients with stable disease after chemotherapy. Healy et al. [37] identified a pre-operative clinical–radiomic model for predicting OS and disease-free survival (DFS) in resectable PDAC patients. In the testing dataset, the model based solely on radiomic features achieved superior results for both OS (C-index of 0.564) and DFS (C-index of 0.573) compared to clinical and AJCC TNM models. In contrast to our findings, the radiomics model outperformed the clinical models. This disparity may be attributed to the exclusion of several clinical features in their models, such as R_margins, alcohol and tobacco consumption. Furthermore, their dataset exclusively included patients who met the criteria for a resectable PDAC.

Regarding the explainability of radiomics features, firstorder_90Percentile (from GTV1) was the unique radiomics features, selected in the OS Clinical&GTV1 model; a lower value showed a higher risk score, meaning that tumours with lower attenuation were associated with a poorer prognosis. This observation aligns with several studies [37,38], indicating that tumours exhibiting reduced enhancement on CT imaging, indicative of decreased vascularity, are associated with poorer OS. Lower density could be related to regions of tissue death due to venous invasion or hypoxia necrosis, both patterns of a more aggressive tumour behaviour [39,40,41,42].

In relation to the explainability of other features in the OS Clinical&GTV1 model, the SHAP analysis proved that high values of CA19, Subgroup, Age, R_Margins and Grading are associated with a higher risk score; it is to be noted that, according to our feature mapping, the highest value for R_Margins represented patients that did not undergo surgery.

With regard to PFS prediction, the SHAP analysis showed a similar trend for Subgroup, CA19, R_Margins and Localization. Investigating the last mentioned feature, the highest value for Localisation meant tumours located at the tail; there are several studies showing different impacts based on the localization of the tumours, which also have contradictory results. Some articles [43,44,45] did not discover a difference in mortality among tumours localised in the head and body/tail; however, other studies [46,47,48] demonstrated that tail or body/tail localisation had a worse prognosis. Moreover, for PFS we observed a trend indicating higher alcohol consumption may lead to a lower risk; as indicated by published research, several studies have presented conflicting outcomes regarding the prognostic significance of alcohol in patients diagnosed with PDAC [49,50,51,52,53].

To summarise, the SHAP analysis reveals and emphasises that, in predicting survival outcomes, clinical features exert a stronger influence compared to radiomics features.

There are several limitations in this study that need to be acknowledged.

The first limitation is the inclusion of data from 49 different centres, resulting in CT exams being performed on various CT scanners with different acquisition protocols. Unfortunately, the utilisation of ComBat for image harmonisation was not feasible due to the unavailability of at least 20 scans per centre, as suggested in [54,55]. To mitigate this bias, we utilised portal phases and resampled the voxel size to a consistent value for all exams.

The second limitation is the statistical differences observed between the training–validation dataset and the testing dataset within patient subgroups (*p*-value of 0.01) and R_margins (*p*-value of 0.02). It is worth considering that the baseline CT scans performed at the HUB IJB and HUB HE (the training–validation datasets) included cases spanning the entire spectrum of severity, and subsequently received surgical or non-surgical treatment options. On the other side, a most important part of exams in the testing dataset (performed in the hospital network connected to HUB) were redirected to the HUB IJB and HUB HE for surgical intervention (without repeating CT exam). This could explain the difference between the two datasets. 

The third limitation is the explainability of radiomics features for temporal data. Radiomics features primarily capture static tumour characteristics, potentially limiting their ability to reflect dynamic changes in disease progression over time. Addressing this concern, delta–radiomics features can be utilised to elucidate the underlying biological mechanisms. Future investigations should aim to incorporate histological and genetic information, enhancing a more accurate representation of temporal changes and boosting the predictive capacity of radiomics models. 

In conclusion, the clinical and radiomics models presented in this study, derived from a multicentric study and designed to predict OS and PFS in patients with PDAC, offer a non-invasive approach with the potential to improve patient management and care. Additionally, we utilised SHAP analysis to gain insights into the interpretability of the model. In order to validate and further enhance the clinical applicability of our findings, it will be mandatory to conduct prospective studies in a real clinical setting and to integrate histological and genetic data.

## 5. Conclusions

Clinical and radiomics models, based on pre-treatment portal CT images, could provide a reliable non-invasive approach for predicting OS and PFS in PDAC, offering an improved strategy for personalised healthcare. The risk score obtained from the best performing models can be applied to stratify patients into high-risk and low-risk groups for both OS and PFS. SHAP analysis demonstrated the particular significance of clinical features in this context. Furthermore, the strength of this study lies in the utilisation of a testing dataset from 47 hospitals, enhancing the generalizability of the proposed models to real clinical practice.

## Figures and Tables

**Figure 1 diagnostics-14-00712-f001:**
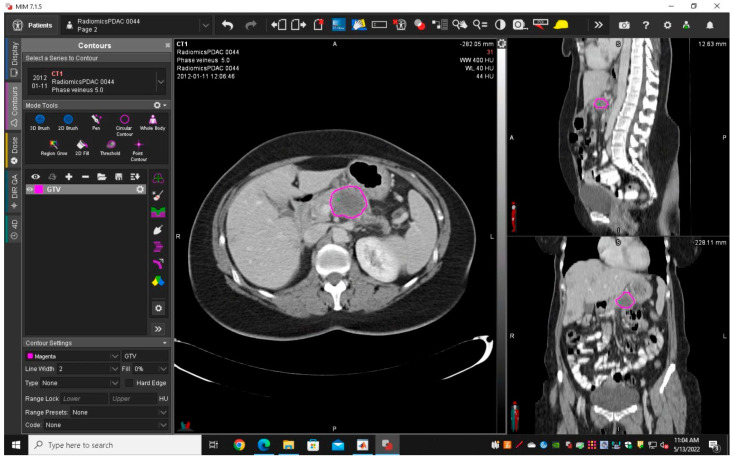
Example of segmentation for GTV1 (purple circle) of a body−localised PDAC using MIM 7.1.5™.

**Figure 2 diagnostics-14-00712-f002:**
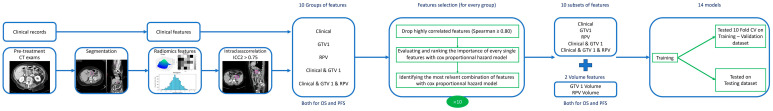
Pipeline of different steps.

**Figure 3 diagnostics-14-00712-f003:**
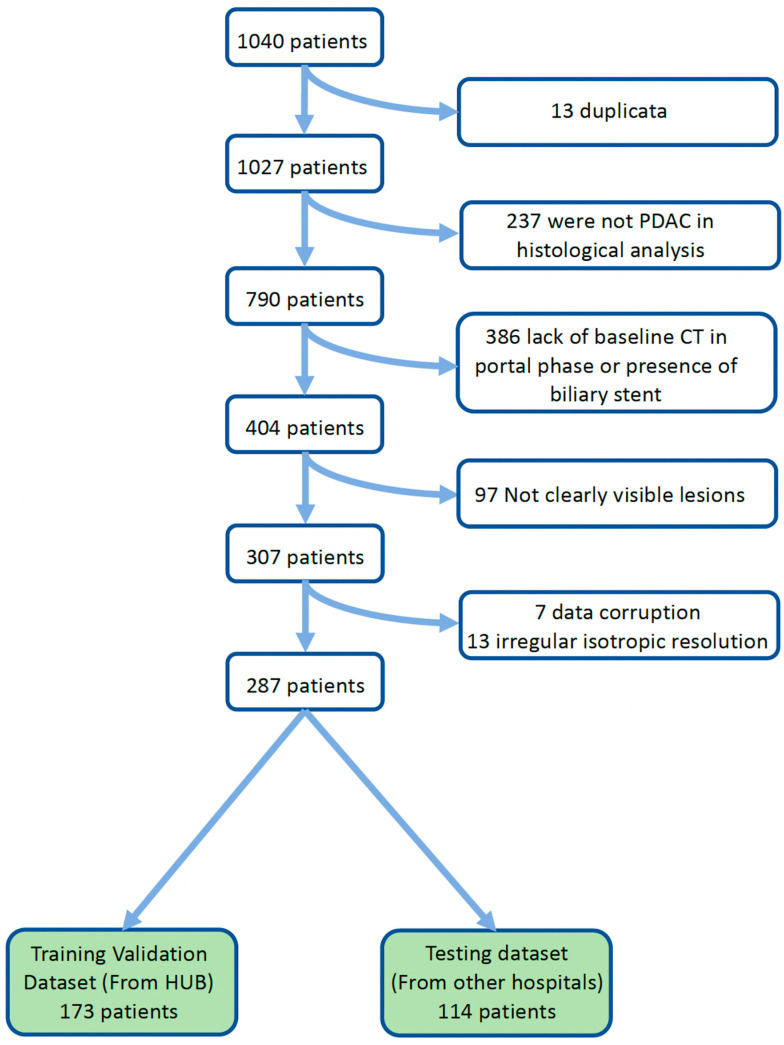
Flowchart of patient inclusion. Appendix A “Patient Exclusion Criteria” illustrates an example of a PDAC case that is not clearly visible, along with details about patients who were excluded due to technical problems.

**Figure 4 diagnostics-14-00712-f004:**
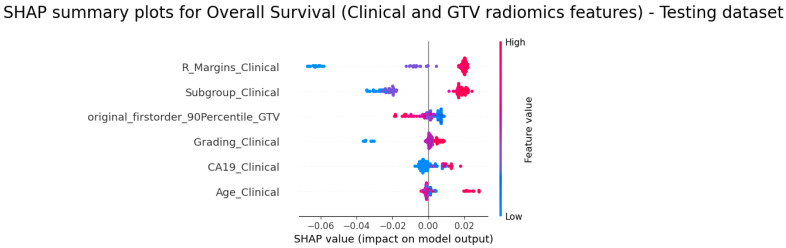
SHAP summary for OS prediction (Clinical&GTV1 model)—testing dataset.

**Figure 5 diagnostics-14-00712-f005:**
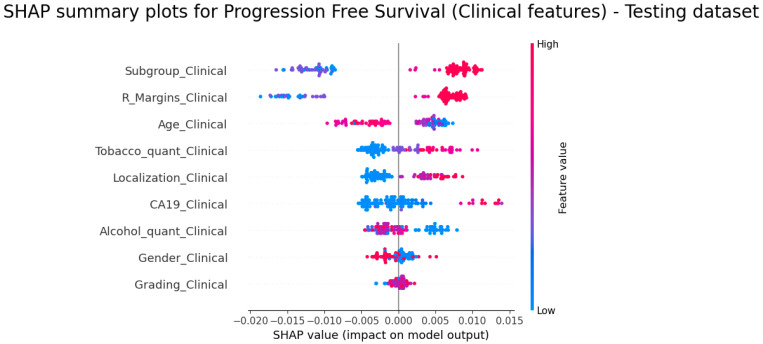
SHAP summary plots for PFS (clinical features)—testing dataset.

**Figure 6 diagnostics-14-00712-f006:**
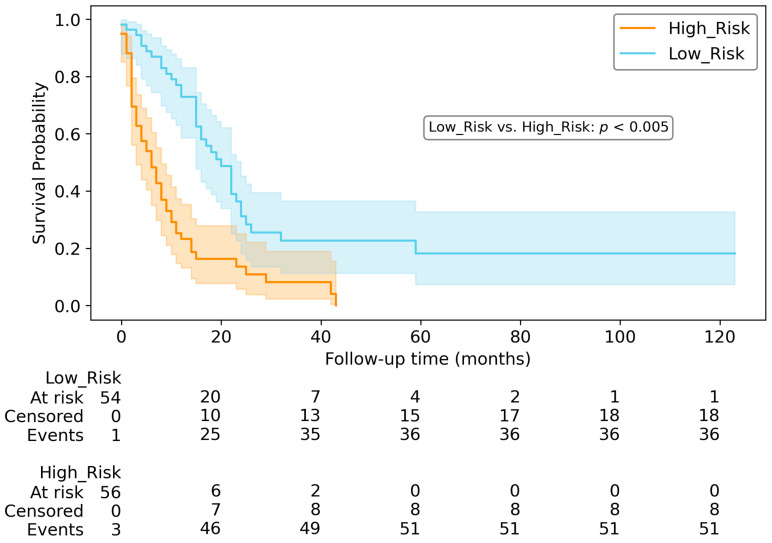
Kaplan–Meyer curve and table counts for OS in testing dataset, using Clinical&GRV1 model. According to the statistical analysis, there was a significant difference in survival between the two groups (*p* < 0.005).

**Figure 7 diagnostics-14-00712-f007:**
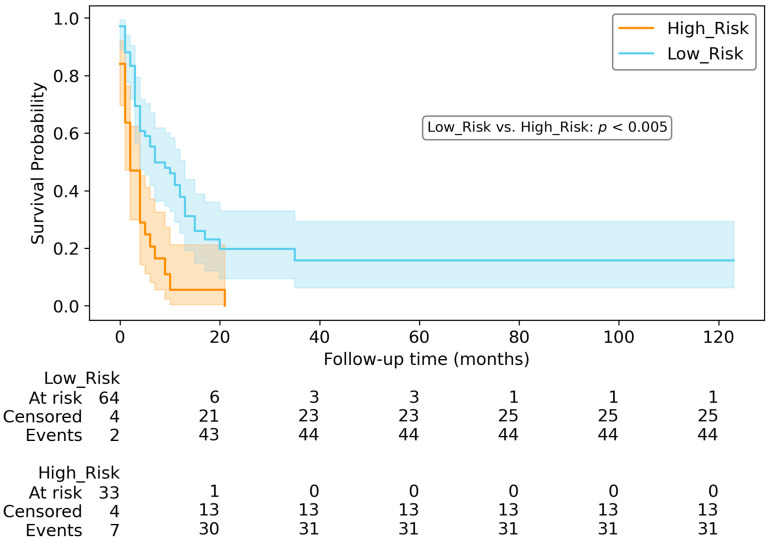
Kaplan–Meyer curve and table counts for PFS in testing dataset, using the Clinical model. According to the statistical analysis, there was a significant difference in survival between the two groups (*p* < 0.005).

**Table 1 diagnostics-14-00712-t001:** Demographic characteristics of the final study population according to the training—validation group and the testing group. Median and range are reported for age, BMI, Ca19.9 and OS. Data for categorical value are presented as: number of observations. PY: pack-year. For treatment type, we refer to the initial treatment approach.

	Datasets 1 and 2	Dataset 3	*p*-Value
Number of patients			
Total	173 (60.3%)	114 (39.7%)	
Age (years)	66 (34–89)	66.5 (38–89)	0.65
Gender			0.77
Men	85 (49.1%)	54 (47.4%)	
Women	88 (50.9%)	60 (52.6%)	
BMI (kg/m^2^)	24.2 (12.84–36)	24.16 (16.76–39.06)	0.72
Tobacco history			0.62
No smoking	79 (45.7%)	57 (50.0%)	
<20 cigarette/day or <15 PY	34 (19.7%)	21 (18.4%)	
>20 cigarette/day or >15 PY	27 (15.6%)	12 (10.5%)	
Stopped	20 (11.6%)	11 (9.6%)	
Alcohol consumption			0.76
No consumption	70 (40.5%)	42 (36.8%)	
<1 unit/day	33 (19.1%)	25 (21.9%)	
1–2 unit/day	16 (9.2%)	14 (12.3%)	
>3 unit/day	26 (15.0%)	14 (12.3%)	
Old consumption	16 (9.2%)	8 (7.0%)	
Ca19.9 (kU/L)	1100 (0.6–243,000)	580 (0.6–357,437)	0.163
Localisation of the lesions			0.04
Head	77 (44.5%)	68 (59.6%)	
Body	57 (33.0%)	26 (22.8%)	
Tail	39 (22.5%)	20 (17.5%)	
Subgroup			0.01
Resectable	13 (7.5%)	16 (14.0%)	
Borderline Resectable	30 (17.3%)	31 (27.2%)	
Unresectable	23 (13.3%)	6 (5.3%)	
Metastatic	107 (61.8%)	61 (53.5%)	
R_margins			0.02
No surgery	144 (83.2%)	79 (69.3%)	
Open-closed without resection	7 (4.0%)	4 (3.5%)	
Resection margin R0	10 (5.8%)	16 (14.0%)	
Resection margin R1-R2	12 (6.9%)	15 (13.2%)	
Histological grade			0.35
Well differentiated	11 (6.4%)	7 (6.1%)	
Moderately differentiated	24 (13.9%)	30 (26.3%)	
Poorly differentiated	31 (17.9%)	25 (21.9%)	
Treatment type			0.24
FOLFIRINOX	57 (33.0%)	46 (40.4%)	
Gemcitabine + Abraxane	85 (49.1%)	48 (42.1%)	
Cisplatin + 5-Fluorouracil	4 (2.3%)	4 (3.5%)	
Gemcitabine + Cisplatin	5 (2.9%)	0 (0.0%)	
No treatment	22 (12.7%)	16 (14.0%)	
Treatment strategy			0.01
Only surgery	2 (1.2%)	8 (7.0%)	
Surgery + Adjuvant	10 (5.8%)	7 (6.1%)	
Neoadjuvant	28 (16.2%)	29 (25.4%)	
Only palliative	133 (77.0%)	70 (61.4%)	
OS (months)	9 (0–72)	10 (0–123)	0.634
PFS (months)	3 (0–28)	3 (0–123)	0.496

**Table 2 diagnostics-14-00712-t002:** Selected Features for predicting models (OS).

Predicting Models for OS				
Clinical	GTV1	RPV	Clinical&GTV1	Clinical&GTV1&RPV
Subgroup	original_firstorder_90Percentile_GTV	original_glcm_ClusterShade_RPV	Subgroup	Subgroup
CA19		original_firstorder_Skewness_RPV	CA19	CA19
R_Margins		original_shape_Sphericity_RPV	R_Margins	R_Margins
Grading		original_glcm_InverseVariance_RPV	Grading	Grading
Age		original_shape_Flatness_RPV	Age	Age
		original_glcm_Contrast_RPV	original_firstorder_90Percentile_GTV	original_glcm_ClusterShade_RPV
		original_shape_LeastAxisLength_RPV		original_firstorder_90Percentile_GTV
				original_firstorder_Skewness_RPV

**Table 3 diagnostics-14-00712-t003:** Feature selected for predicting models (PFS).

Predicting Models for PFS				
Clinical	GTV1	RPV	Clinical&GTV1	Clinical&GTV1&RPV
Subgroup	original_glcm_Contrast_GTV	original_glcm_Contrast_RPV	Subgroup	Subgroup
CA19		original_glcm_Correlation_RPV	CA19	CA19
R_Margins			R_Margins	original_glcm_Correlation_RPV
Localization			original_glcm_Contrast_GTV	R_Margins
Age			original_firstorder_90Percentile_GTV	
Alcohol_quant			original_glcm_Correlation_GTV	
Gender			Localization’	
Tobacco_quant			original_glcm_Idn_GTV	
Grading			Age	

**Table 4 diagnostics-14-00712-t004:** The C-index results for the training–validation and testing datasets for OS (* indicates a statistically significant difference between the proposed model and the clinical model).

Models	Training–Validation Set	Testing Set	*p*-Value
CLINICAL	0.71 ± 0.07	0.7	-
GTV1	0.55 ± 0.05	0.63	0.156
RPV	0.62 ± 0.06	0.56	0.001 *
CLINICAL&GTV1	0.72 ± 0.08	0.72	0.212
CLINICAL&GTV1&RPV	0.71 ± 0.08	0.71	0.817
GTV1 volume	0.5 ± 0.08	0.58	0.005 *
RPV volume	0.5 ± 0.1	0.44	0.0002 *

**Table 5 diagnostics-14-00712-t005:** The C-index results for the training–validation and testing datasets for PFS (* indicates a statistically significant difference between the proposed model and the clinical model).

Models	Training–Validation Set	Testing Set	*p*-Value
CLINICAL	0.67 ± 0.08	0.7	-
GTV1	0.62 ± 0.08	0.51	0.004 *
RPV	0.55 ± 0.09	0.5	0.002 *
CLINICAL&GTV1	0.67 ± 0.11	0.66	0.657
CLINICAL&GTV1&RPV	0.63 ± 0.06	0.67	0.388
GTV1 volume	0.52 ± 0.08	0.51	0.001 *
RPV volume	0.48 ± 0.06	0.54	0.003 *

## Data Availability

The Python functions used for feature selection, model training, and model evaluation are available on GitHub at the following repository: https://github.com/roberto-casale/RadPanc-clinical-radiomics-based-models (accessed on 13 January 2024).

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
