# Peer review of "Development of Clinical Radiomics-Based Models to Predict Survival Outcome in Pancreatic Ductal Adenocarcinoma: A Multicenter Retrospective Study"

_diagnostics, 2024, doi:10.3390/diagnostics14070712_

Round 1
Reviewer 1 Report
Comments and Suggestions for Authors
I would like to express that I read your study titled "Development of clinical radiomics-based models to predict survival outcome in pancreatic ductal adenocarcinoma: a multicenter retrospective study" in detail and took great pleasure in reading it. To evaluate the study in general, it can be seen that the Abstract section is well written, but especially the subheadings are not detailed enough. The concept of HUB, which is frequently mentioned in the Abstract section, should be written in detail before abbreviating it. At the end of the introduction, a paragraph about the article's contributions and innovations to the literature should be added, and after that, a paragraph about the organization of the article should be included. The section written at the top is the material and method section. I would like to comment generally on the shortcomings in this section. For example, how was feature extraction done, which methods were used, which methods were used in feature selection, how many features were selected, and how many features were eliminated? Unfortunately, I would like to point out that there are no answers to these questions. Similarly, I think the word radiomics is only mentioned in the title. It should be stated where and how these methods are applied. At this stage, it is necessary to add a flow diagram or figure for the proposed model. This will help the reader understand the article better. Figures 2,5,6 definitely need to be redrawn. Visually it looks very rough. It is not clear how you use the radiomics data together with the clinical data in Table 1. I understand that it was used in the Abstract, but I couldn't find it afterward. There is no discussion section in the article. I believe that if the entire article is written like the Abstract section, a very good article will emerge.
Comments on the Quality of English LanguageThere are many spelling and grammatical errors in the paper. The article must be reviewed by researchers.
Author Response
I would like to express that I read your study titled "Development of clinical radiomics-based models to predict survival outcome in pancreatic ductal adenocarcinoma: a multicenter retrospective study" in detail and took great pleasure in reading it. To evaluate the study in general, it can be seen that the Abstract section is well written, but especially the subheadings are not detailed enough.
- Thank you for your feedback. We have reviewed and revised the highlights and introduction section to provide a more detailed and specific overview of our study. We hope that these adjustments improve the manuscript's clarity and informativeness.
The concept of HUB, which is frequently mentioned in the Abstract section, should be written in detail before abbreviating it.
- Following your recommendation, we have amended the text to introduce the full term associated with the abbreviation "HUB" prior to its first usage. We hope this change ensures clarity and aids readers in comprehending the terminology used throughout the document.
At the end of the introduction, a paragraph about the article's contributions and innovations to the literature should be added, and after that, a paragraph about the organization of the article should be included.
- Thank you for your suggestion to elaborate on the contributions, innovations, and organization of our article. We have added a paragraph at the end of the introduction highlighting the study's contributions, particularly its multicenter approach and the evaluation of its relevance in diverse clinical settings, also through the analysis of both clinical and radiomic features within the predictive models. We have also included a brief outline of the article's structure to guide the readers through our manuscript.
The section written at the top is the material and method section. I would like to comment generally on the shortcomings in this section. For example, how was feature extraction done, which methods were used, which methods were used in feature selection, how many features were selected, and how many features were eliminated? Unfortunately, I would like to point out that there are no answers to these questions.
- We have revised the section “2.5. Feature selection” to offer a detailed exposition of our approach. Specifically, we conducted feature selection on both clinical data and CT radiomics features, categorizing them into distinct groups for analysis concerning overall survival and progression-free survival. The initial phase involved identifying the most reliable features via the IntraClass Correlation coefficient. Subsequently, within each group, we eliminated redundant features using a threshold based on the Spearman correlation coefficient. Following this, the predictive value of each feature was evaluated through univariate analysis utilizing the Cox Proportional Hazards model. A methodical process was then employed to ascertain the most pertinent combination of features, again leveraging the Cox Proportional Hazards model, with the C-index serving as the performance metric.
Further details are available in our GitHub repository shared in the main text.
Similarly, I think the word radiomics is only mentioned in the title. It should be stated where and how these methods are applied. At this stage, it is necessary to add a flow diagram or figure for the proposed model. This will help the reader understand the article better.
- We have revised the text, in particular below the section “2.3. Image acquisition” and section “2.5. Feature selection” to include a more detailed description of how radiomic analysis is integrated into our study post the initial CT scan phase. Specifically, we have elucidated the process of feature extraction from CT images and how these features are employed alongside clinical data to construct our predictive models. We acknowledge the suggestion to add a figure to enhance readers' comprehension of the methodological workflow. Lastly, we have added Figure 2 to show the pipeline of the model building and of the different steps of the study.
Figures 2,5,6 definitely need to be redrawn. Visually it looks very rough.
- We have made enhancements to these figures to ensure they are presented with the utmost clarity. In particular, we have redrawn the Kaplan-Meier figures by implementing several improvements (figure resolution to 300 DPI to ensure that the images are sharp and clear when printed or viewed on screen; we adjusted the font size and grid layout to enhance readability and ensure that all text and markers are easily discernible).
It is not clear how you use the radiomics data together with the clinical data in Table 1. I understand that it was used in the Abstract, but I couldn't find it afterward.
- We acknowledge the necessity for clearer exposition on this matter and have therefore revised the text, specifically sections 2.4 "Segmentation and Feature Extraction" and 2.5 "Feature Selection", to elucidate how radiomics and clinical data were synergistically utilized in our study. In the revised sections, we detail the process of extracting radiomic features from CT and describe how these features were then combined with clinical data to enhance the predictive models. We clarify the methodology employed to integrate these two data types.
There is no discussion section in the article. I believe that if the entire article is written like the Abstract section, a very good article will emerge.
- We have revised and enhanced the discussion. The discussion delves into the implications of our research, compares our results with existing literature, outlines limitations and future directions.
Reviewer 2 Report
Comments and Suggestions for Authors
-
1. What criteria guided the authors in choosing the 107 features? Was there a rationale for not opting for a smaller or larger set of features?
-
2. There are concerns about the models' performance, particularly noting a generally low C-index.
-
3. What motivated the author to utilize the XGGradientBoostingSurvivalAnalysisBoost classifier? Has there been a comparative analysis with other models?
Editing is required.
Author Response
1. What criteria guided the authors in choosing the 107 features? Was there a rationale for not opting for a smaller or larger set of features?
- The features extracted are categorized into 3D shape, first-order, GLCM, GLDM, GLRLM, GLSZM, and NGTDM classes. This selection is aligned with the standard feature extraction protocol as delineated in the PyRadiomics documentation (https://pyradiomics.readthedocs.io/en/latest/features.html), which standardly extracts 107 features—specifically, 14 shape 3D, 18 first order, 24 GLCM, 14 GLDM, 16 GLRLM, 16 GLSZM, and 5 NGTDM features.
It's important to note that the total number of features that could be extracted is 110. However, three particular features (first-order Standard Deviation, 3D shape Compactness 1, 3D shape Compactness 2, and 3D shape Spherical Disproportion) are disabled by default during the extraction process. This is because they are highly correlated with other normally extracted features, and excluding them helps in minimizing redundancy and potential multicollinearity in the feature set.
This standardized approach to feature extraction is designed to ensure a comprehensive yet relevant dataset for our analysis, avoiding the inclusion of redundant features while maintaining a robust feature set for model development.
According to this point and for clarity, we have included it in the text (2.4. Segmentation and features extraction) the exact number of features for each group.
2. There are concerns about the models' performance, particularly noting a generally low C-index.
- Thank you for addressing your concerns regarding the performance of our models, particularly the observed C-index values. According to the scikit-survival 0.22.2 manual (https://scikit-survival.readthedocs.io/en/stable/user_guide/random-survival-forest.html), a concordance index of 0.68 is considered a good value; in the link is reported also a reference for supporting this data (DOI: 10.1214/08-AOAS169). Furthermore, other studies in the literature (such as the one found at DOI: 10.1371/journal.pone.0214551; DOI: 10.3390/cancers14051228; DOI: 10.1016/j.hpb.2020.07.007), also report similar C-index values, further substantiating the relevance and reliability of our models' performance metrics. These comparisons underscore that our models' C-index values are within the accepted range for predictive models in survival analysis, affirming their potential utility.
3. What motivated the author to utilize the XGGradientBoostingSurvivalAnalysisBoost classifier? Has there been a comparative analysis with other models?
- Our decision was informed by the flexibility and effectiveness of the gradient boosting (XGGradientBoostingSurvivalAnalysisBoost) framework, as detailed in the scikit-survival 0.22.2 manual. Gradient boosting is not confined to a single model but is a versatile framework that optimizes various loss functions. Another alternative is Random Survival Forest. However, while Random Survival Forest is a viable alternative, employing multiple Survival Trees independently and averaging their predictions, gradient boosting constructs its model sequentially, which can lead to more refined and potentially more accurate predictions. This sequential construction influenced our decision to discard Random Survival Forest and to opt for the Gradient Boosting classifier, given its potential for optimization and adaptability.
Additionally, the literature supports the utility of Gradient Boosting classifiers (e.g.: DOI: 10.3390/cancers15071932)
Round 2
Reviewer 1 Report
Comments and Suggestions for Authors
I would like to thank the researchers for this successful revision.